# Membranes prime the RapGEF EPAC1 to transduce cAMP signaling

Candice Sartre[1,6], François Peurois[1,6], Marie Ley [2], Marie-Hélène Kryszke [1], Wenhua Zhang[1], Delphine Courilleau[3], Rodolphe Fischmeister [4], Yves Ambroise[5], Mahel Zeghouf [1], Sarah Cianferani [2], Yann Ferrandez[1] & Jacqueline Cherfils [1] ✉

EPAC1, a cAMP-activated GEF for Rap GTPases, is a major transducer of cAMP signaling and a therapeutic target in cardiac diseases. The recent discovery that cAMP is compartmentalized in membrane-proximal nanodomains challenged the current model of EPAC1 activation in the cytosol. Here, we discover that anionic membranes are a major component of EPAC1 activation. We find that anionic membranes activate EPAC1 independently of cAMP, increase its affinity for cAMP by two orders of magnitude, and synergize with cAMP to yield maximal GEF activity. In the cell cytosol, where cAMP concentration is low, EPAC1 must thus be primed by membranes to bind cAMP. Examination of the cell-active chemical CE3F4 in this framework further reveals that it targets only fully activated EPAC1. Together, our findings reformulate previous concepts of cAMP signaling through EPAC proteins, with important implications for drug discovery.

The second messenger, cyclic adenosine monophosphate (cAMP), transduces extracellular signals conveyed by hormones and neurotransmitters through G protein coupled receptors and heterotrimeric G proteins. In cells, the opposite activities of G protein-regulated adenylyl cyclases, which synthesize cAMP, and phosphodiesterases, which degrade it, result in cAMP fluxes that regulate a broad array of major intracellular pathways in physiological and pathological conditions (reviewed in refs. 1, 2). While cAMP has long been viewed as a freely diffusible molecule, it is now widely recognized that it is instead segregated into membrane-proximal nanometer-sized compartments, thus maintaining its concentration low in the bulk cytosol and warranting the specificity of the cellular responses (refs. 3–6; reviewed in ref. 7). When its concentration increases, cAMP binds to, and activates, specific targets, of which protein kinase A (PKA) is the best characterized (reviewed in ref. 8). Another major transducer of cAMP signaling is the EPAC protein family, comprised of EPAC1 and EPAC2. EPAC proteins function as guanine nucleotide exchange factors (GEFs)

to activate small GTPases of the Ras-related Rap family independently of PKA[9,10]. In humans, EPAC proteins regulate essential functions, such as integrin-mediated cell adhesion and cell–cell junction formation, and control important physiological processes such as insulin secretion, neurotransmitter release, and cardiac functions (reviewed in refs. 11–13). Animal and clinical studies have implicated EPAC1 and/or EPAC2 in diverse human diseases, notably cancer (reviewed in ref. 14), chronic inflammatory pain[15], pathological angiogenesis[16], and heart diseases (reviewed in refs. 12, 17). EPAC proteins are, therefore, highly studied as pharmacological targets for therapeutic intervention (reviewed in refs. 13, 18).

An important step towards targeting EPAC proteins by small molecules in diseases is to understand the molecular basis for their regulation. Seminal biochemical and structural studies uncovered how cAMP controls the activation of Rap GTPases by EPAC proteins[10,19,20]. In vitro, EPAC1 and EPAC2 are strongly autoinhibited in the absence of cAMP[10]. The crystal structure of full-length, autoinhibited

[1]Université Paris-Saclay, Ecole Normale Supérieure Paris-Saclay, CNRS, 91190 Gif-sur-Yvette, France. [2]Laboratoire de Spectrométrie de Masse BioOrganique, Université de Strasbourg, IPHC, CNRS UMR 7178, Infrastructure Nationale de Protéomique ProFI – FR2048, 67087 Strasbourg, France. [3]Université Paris-Saclay, IPSIT-CIBLOT, Inserm US31, CNRS UAR3679, 91400 Orsay, France. [4]Université Paris-Saclay, INSERM, UMR-S 1180, 91400 Orsay, France. [5]Université Paris-Saclay, CEA, Service de Chimie Bioorganique et de Marquage, 91191 Gif-sur-Yvette, France. [6]These authors contributed equally: Candice Sartre, François Peurois. ✉e-mail: jacqueline.cherfils@ens-paris-saclay.fr

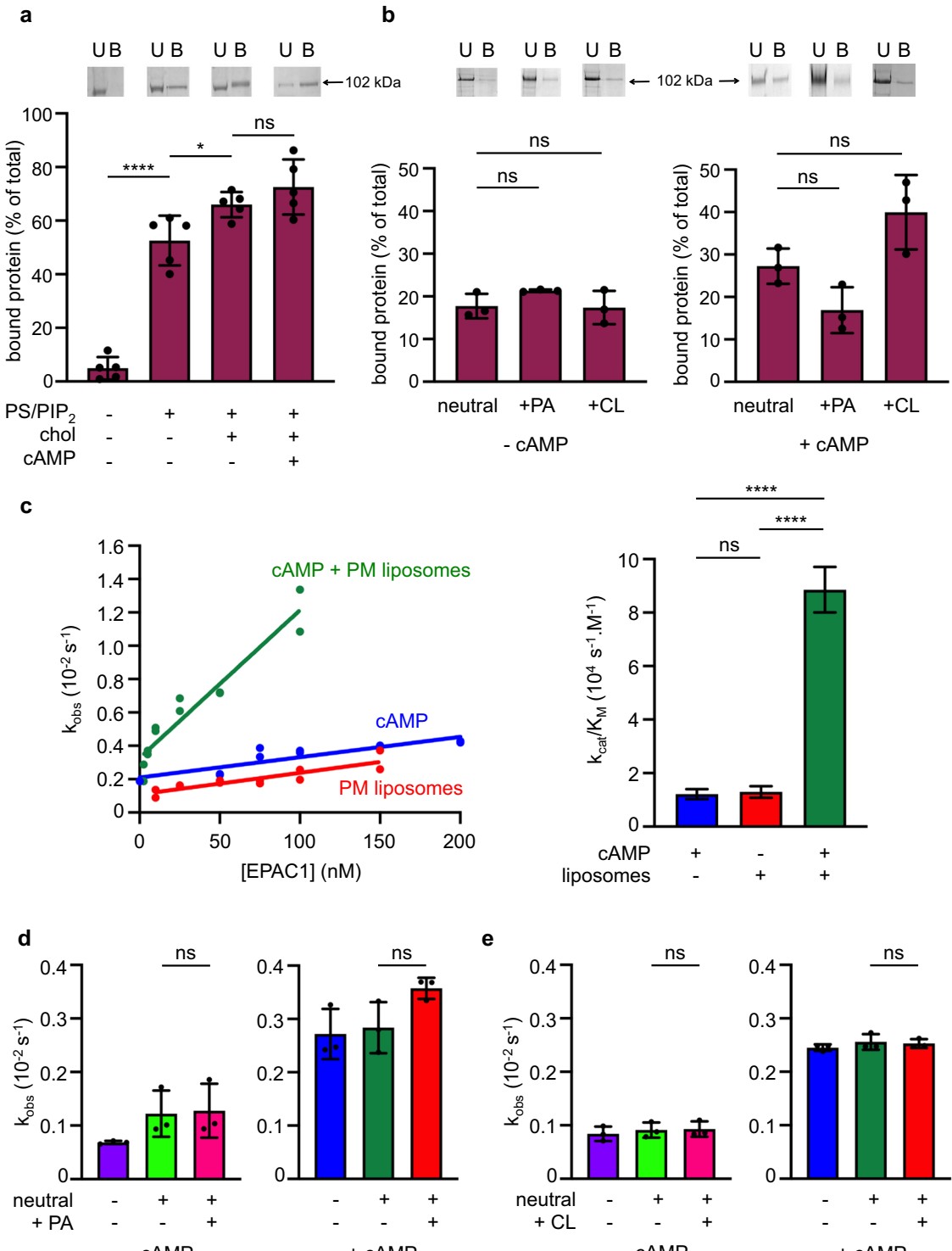

**Fig. 1 | Anionic membranes activate EPAC1^FL. a, b** EPAC1^FL binding to PC/PE (neutral) liposomes, supplemented or not with PIP$_2$/PS ± cholesterol (**a**), or with PA or CL (**b**), in the presence or absence of cAMP, was analyzed by flotation. Top, representative Coomassie blue staining of bound (B) and unbound (U) protein. Bottom, diagrams showing % bound protein as means ± SD of $n = 5$ (**a**) or $n = 3$ (**b**) independent experiments, analyzed by one-way ANOVA followed by Sidak's (**a**) or Dunnett's (**b**) multiple comparison test, providing adjusted $P$ values. **c** EPAC1^FL activity in solution with cAMP (blue), or on PM liposomes with (green) or without cAMP (red) as a function of enzyme concentration, modeled by nonlinear regression (straight line), with all points considered independent (left). The slopes of the curves deduced from the regressions were reported as $k_{cat}/K_M$ ± SE ($n = 1$ experiment including 12 individual kinetics reactions per condition) and analyzed by Brown−Forsythe ANOVA, followed by a Dunnett's T3 multiple comparison test, providing adjusted $P$ values (right). **d, e** EPAC1^FL nucleotide exchange activity was measured in solution, or on neutral liposomes supplemented or not with PA (**d**) or CL (**e**), in the presence or absence of cAMP. The results are presented as means ± SD of $n = 3$ independent experiments and analyzed by two-tailed unpaired Student's $t$- tests. **a−e** ns: $P > 0.05$, not significant; *$P < 0.05$; **$P < 0.01$; ****$P < 0.0001$. Test statistics, mean values with 95% CI, effect sizes, and exact $P$ values are in Supplementary Tables 4, 5. Source data are provided as a Source Data file.

EPAC2 showed that its N-terminal regulatory region, which is comprised of a cyclic nucleotide-binding (CNB) domain, a Disheveled/Egl-10/Pleckstrin (DEP) domain and a second CNB domain, hinders access to the Rap-binding site located in the C-terminal GEF domain[19]. No structure of an activated full-length EPAC protein is currently known. However, the crystal structure of a truncated EPAC2 construct bound to a cAMP analog and Rap1 showed that cAMP locks an alternative CNB-GEF intramolecular interface, thereby removing the CNB domain from the GEF active site through a large rigid-body movement[20]. EPAC1 has the same domain organization as EPAC2, apart from the replacement of the first CNB domain by a 50-residue domain without recognizable structural homology, suggesting that its mechanism of autoinhibition release shares common features with that of EPAC2.

Biochemical analyses suggest, however, that the regulation of EPAC proteins by cAMP is more complex than a simple binary conformational switch. Notably, the dissociation constant of EPAC1 for cAMP is in the 3–4 μM range[21–24], and half-maximal activation is achieved in the 30–45 μM range[23,25,26]. Such concentrations are considerably higher than those reported for PKA[24,25], and they may not be attained in the bulk cytosol where inactive EPAC proteins are located. Furthermore, cAMP is unable to yield full activation of EPAC1 in vitro even at saturating concentration, as shown by its comparison to the EPAC-specific cAMP analog 8-(4-chlorophenylthio)−2'-O-methyl-cAMP (8CPT-2Me-cAMP, also called 007), which yields a maximal activity three times higher than that achieved by cAMP[25–28]. Thus, EPAC proteins may have activation determinants other than cAMP, which remain to be identified.

The pivotal role of membranes in regulating the function of small GTPases (reviewed in ref. 29) and their regulators (reviewed in refs. 30, 31) is currently gaining center stage in small GTPase biology. Like most, if not all, small GTPase/GEF systems, EPAC and Rap proteins actuate their functions at the surface of membranes. Rap GTPases associate reversibly with membranes through their prenylated C-terminal extension[32–35]. Likewise, ectopically expressed EPAC1 translocates from the cytosol to the plasma membrane upon an increase in cAMP levels[36,37]. How EPAC1 is recruited to the membrane has, however, remained poorly understood. Mutagenesis studies of EPAC1 in vitro and in cells suggested that it binds directly to the membrane through a polybasic loop of the DEP domain, a domain often involved in membrane binding, and binding was proposed to be specific for phosphatidic acid lipids (PA)[37,38]. Hydrogen/deuterium exchange coupled to mass spectrometry (HDX-MS) of EPAC2 suggested that this loop becomes solvent-exposed upon activation by cAMP[39], yet an isolated DEP domain, in which this loop is exposed, did not locate to the plasma membrane[36]. How activation by cAMP and membrane recruitment are coupled thus remains to be elucidated.

Small molecules that interfere with complex regulatory mechanisms can bring decisive insight into elusive functional determinants, as exemplified by Brefeldin A, a natural compound that highlighted the mechanism of activation of Arf GTPases by their GEFs at the interface with membranes[40]. Several agonists and inhibitors of EPAC1 and EPAC2 have been described (reviewed in refs. 13, 18, 41–43). In that regard, the EPAC1-specific inhibitor CE3F4 is especially interesting for its intriguing inhibitory properties. CE3F4 was discovered from a fluorescence-based GEF screen using a recombinant EPAC1 construct containing the CNB and GEF domains[44]. It efficiently inhibits the activation of the Rap1 GTPase in cardiomyocytes[44] and prevents arrhythmia in mice[45] and rats[46], thus showing promising potential for drug development in cardiac pathologies. Surprisingly, CE3F4 does not inhibit the GEF domain of EPAC1 alone[44], and NMR studies showed that it binds to the isolated CNB domain without competing with cAMP[47]. Thus, CE3F4 appears to act through an allosteric mechanism with respect to both cAMP and Rap GTPases, yet to be elucidated, which makes it an appealing tool to interrogate the mechanism of EPAC1 activation.

In this study, we characterized the interplay between cAMP and membranes in the activation of EPAC1 by combining reconstitutions of purified proteins in artificial membranes, quantitative fluorescence kinetics, structural analysis by crystallography, small angle X-ray scattering (SAXS) and HDX-MS and inhibition studies using CE3F4. Our data reveal a major role of membranes in cAMP and EPAC signaling, with important implications for the activation of EPAC1 in cells, for cAMP signaling, and for drug discovery.

## Results
### Membranes activate EPAC1 in the absence and presence of cAMP
To determine whether membranes affect EPAC1 activity directly, we used purified full-length EPAC1 (EPAC1$^{FL}$ hereafter) (Supplementary Fig. 1a, b) and liposomes of controlled lipid composition. First, we characterized the interaction of EPAC1$^{FL}$ with membranes, using a liposome flotation assay (Fig. 1a). EPAC1$^{FL}$ did not bind to liposomes containing only phosphatidylcholine (PC) and phosphatidylethanolamine (PE) lipids (neutral liposomes hereafter). In contrast, strong binding to liposomes containing phosphatidylserine (PS) and phosphatidylinositol-4,5-bisphosphate (PIP$_2$) was observed. Binding was slightly increased by enrichment of PS-PIP$_2$ liposomes with cholesterol, while the addition of cAMP in the flotation assay had no discernible effect. Next, we characterized the interaction of EPAC1 with phosphatidic acid (PA), which has been reported to drive the localization of EPAC1 to the plasma membrane[37], and with cardiolipin (CL), which is enriched in cardiomyocyte mitochondria (reviewed in ref. 48), an organelle where a subpopulation of EPAC1 has been reported to carry out important functions[49,50], using PC-PE liposomes enriched with 10% PA (PA-liposomes hereafter) or 10% CL (CL-liposomes hereafter) (Fig. 1b). Surprisingly, we observed no increase in EPAC1$^{FL}$ binding to PA- or CL-liposomes compared to neutral liposomes. Likewise, the addition of cAMP did not increase the binding of EPAC1$^{FL}$ to PA- or CL- liposomes compared to neutral liposomes.

Next, we analyzed whether the binding of EPAC1 to membranes modulates its GEF activity towards the small GTPase Rap1, using liposomes containing PS, PIP$_2$, and cholesterol (plasma membrane (PM) liposomes hereafter) to which EPAC1$^{FL}$ binds strongly. The efficiency of EPAC1-stimulated GDP/GTP exchange was monitored by following the kinetics of dissociation of the fluorescent nucleotide BODIPY-GDP from Rap1. We used non-lipidated full-length Rap1A (Rap1 hereafter), which binds to PM liposomes (Supplementary Fig. 1c) likely through its polybasic C-terminal region ($^{167}$RKTPVEKKKPKKKSCLLL$^{184}$), thus allowing to reconstitute the membrane-associated reaction. $k_{obs}$ were measured over a range of EPAC1 concentrations and used to determine catalytic efficiencies ($k_{cat}/K_M$, expressed in s$^{-1}$ M$^{-1}$) (Fig. 1c and Supplementary Fig. 1d). In solution, EPAC1$^{FL}$ had no GEF activity towards Rap1 in the absence of cAMP (Supplementary Fig. 1d). Addition of cAMP supported modest GEF activity ($k_{cat}/K_M = 1.2 \times 10^4$ s$^{-1}$ M$^{-1}$), thus confirming that EPAC1$^{FL}$ autoinhibition is released by cAMP. Unexpectedly, PM liposomes alone were equally efficient at activating EPAC1$^{FL}$ as cAMP alone ($k_{cat}/K_M = 1.3 \times 10^4$ s$^{-1}$ M$^{-1}$). Remarkably, the addition of both cAMP and PM liposomes resulted in a marked increase in GEF efficiency as compared to either cAMP or PM liposomes alone ($k_{cat}/K_M = 8.9 \times 10^4$ s$^{-1}$ M$^{-1}$). PA-liposomes (Fig. 1d and Supplementary Fig. 1e) and CL-liposomes (Fig. 1e and Supplementary Fig. 1f) showed no increase in activity compared to neutral liposomes with or without cAMP, consistent with the lack of interaction of EPAC1$^{FL}$ with these liposomes.

We conclude from these observations that EPAC1 binds strongly to PM liposomes but does not recognize PA or CL lipids directly, that PM liposomes are able to activate EPAC1 in the absence of cAMP, and that cAMP and PM liposomes synergize to yield maximal EPAC1 activity.

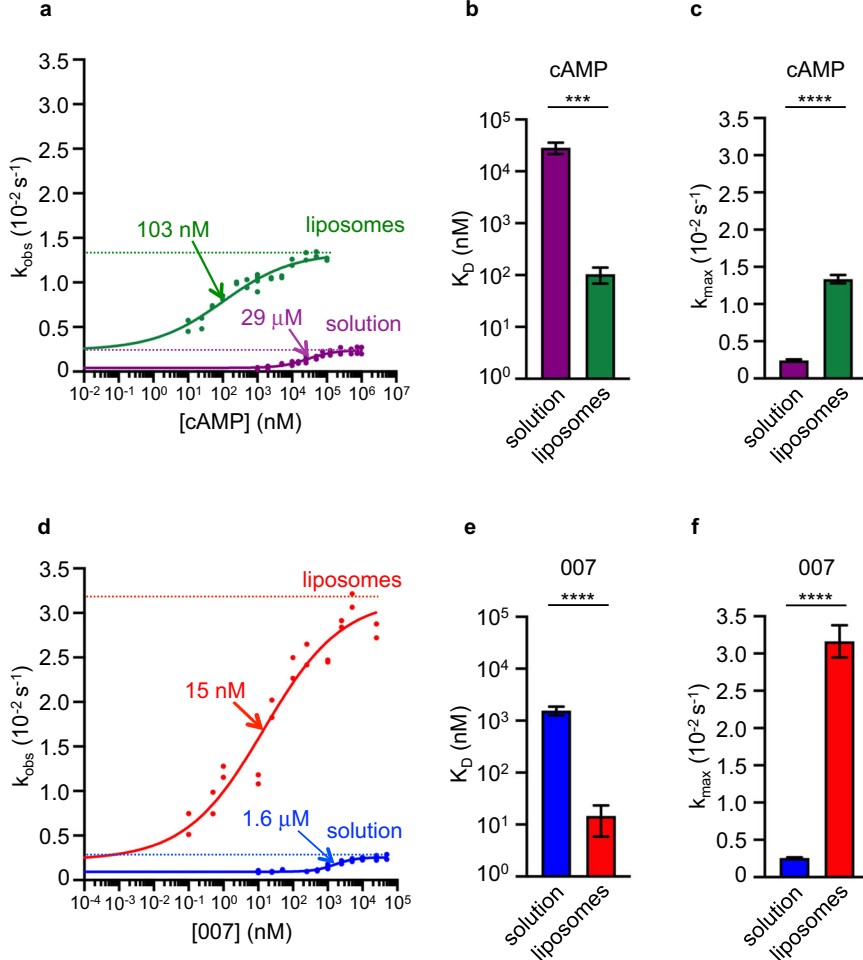

**Fig. 2 | Membranes increase the affinity of EPAC1$^{FL}$ for cAMP and 007.**
**a**, **d** EPAC1$^{FL}$ nucleotide exchange activity was measured in the presence of different concentrations of cAMP (**a**) in solution (purple) or on liposomes (green), or in the presence of different concentrations of 007 (**d**) in solution (blue) or on liposomes (red). Sigmoidal curves were fitted to the data using nonlinear regression, with all points considered independent. EC50 (arrows) and top values (dotted lines) of the curves were deduced from the regression and reported as $K_D \pm$ SE (**b**, **e**) and

$k_{max} \pm$ SE (**c**, **f**), respectively ($n = 1$ experiment including 32 (cAMP), 28 (007 in solution), or 24 (007 on liposomes) individual kinetics reactions per condition). Statistical analyses were performed by two-tailed unpaired Student's $t$-tests with Welch's correction. ***$P < 0.001$; ****$P < 0.0001$. Test statistics, mean values with 95% CI, effect sizes, and exact $P$ values are in Supplementary Table 5. Source data are provided as a Source Data file.

## The affinity of EPAC1 for cAMP is dramatically increased by membranes

The synergy between cAMP and membranes suggests that membranes could affect how cAMP binds to EPAC1. We determined the apparent dissociation constant ($K_D$) of cAMP in solution and in the presence of PM liposomes (Fig. 2a, b), by monitoring the kinetics of Rap1 activation by EPAC1$^{FL}$ as a function of cAMP concentration. The apparent $K_D$ was determined as the concentration of cAMP yielding half maximum stimulation of EPAC1 activity, as previously described for EPAC proteins in solution[26,28]. In solution, the apparent $K_D$ was 29 μM, a value that is even higher than the mediocre $K_D$ determined previously for truncated EPAC1 constructs using this and other methods[21,26,28]. Remarkably, the apparent $K_D$ decreased to 103 nM in the presence of PM liposomes, representing a 278-fold increase in affinity. PM liposomes also increased the maximal activity ($k_{max}$) by sixfold, from $0.24 \times 10^{-2}$ s$^{-1}$ in solution to $1.33 \times 10^{-2}$ s$^{-1}$ (Fig. 2a, c).

The widely used cAMP super agonist 007 has been reported to increase $k_{max}$ by threefold in solution compared to cAMP[25], which is in the same range as the increase of $k_{max}$ promoted by membranes in the presence of cAMP. We, therefore, asked whether activation of EPAC1$^{FL}$ by 007 remains sensitive to membranes, using the same approach as above (Fig. 2d–f). The apparent $K_D$ of 007 in solution was 1.6 μM, a

value that is in the same range as previously reported in refs. 26, 28. Surprisingly, PM liposomes decreased the $K_D$ of 007 to 15 nM, representing a 107-fold increase in affinity. PM liposomes also increased $k_{max}$ from $0.26 \times 10^{-2}$ s$^{-1}$ in solution to $3.16 \times 10^{-2}$ s$^{-1}$, representing a 12-fold increase. Thus, activation of EPAC1 by 007 is strongly potentiated by the presence of membranes.

We conclude that anionic liposomes strongly potentiate the capability of cAMP to activate EPAC1, and that both the $K_D$ and $k_{max}$ of 007 surpass those of cAMP in the context of membranes.

### Structural determinants of EPAC1 binding to membranes

The DEP domain is the only canonical membrane-binding domain in EPAC1 (reviewed in ref. 38). To gain insight into the membrane-binding site of EPAC1, we first determined the crystal structure of a construct encompassing the DEP and CNB domains (EPAC1$^{DEP-CNB}$, residues 50–318) in complex with cAMP at 2.3 Å resolution (Fig. 3a, crystallographic statistics in Supplementary Table 1). The structure shows that the DEP and CNB domains are connected by a kinked helix (residues 154–181, with the kink located at P171), which is sandwiched between the two domains and entirely mediates their interaction. This organization is similar to that seen in the structure of unbound EPAC2$^{CNB-DEP}$[51] and of autoinhibited full-length EPAC2[19], with only a

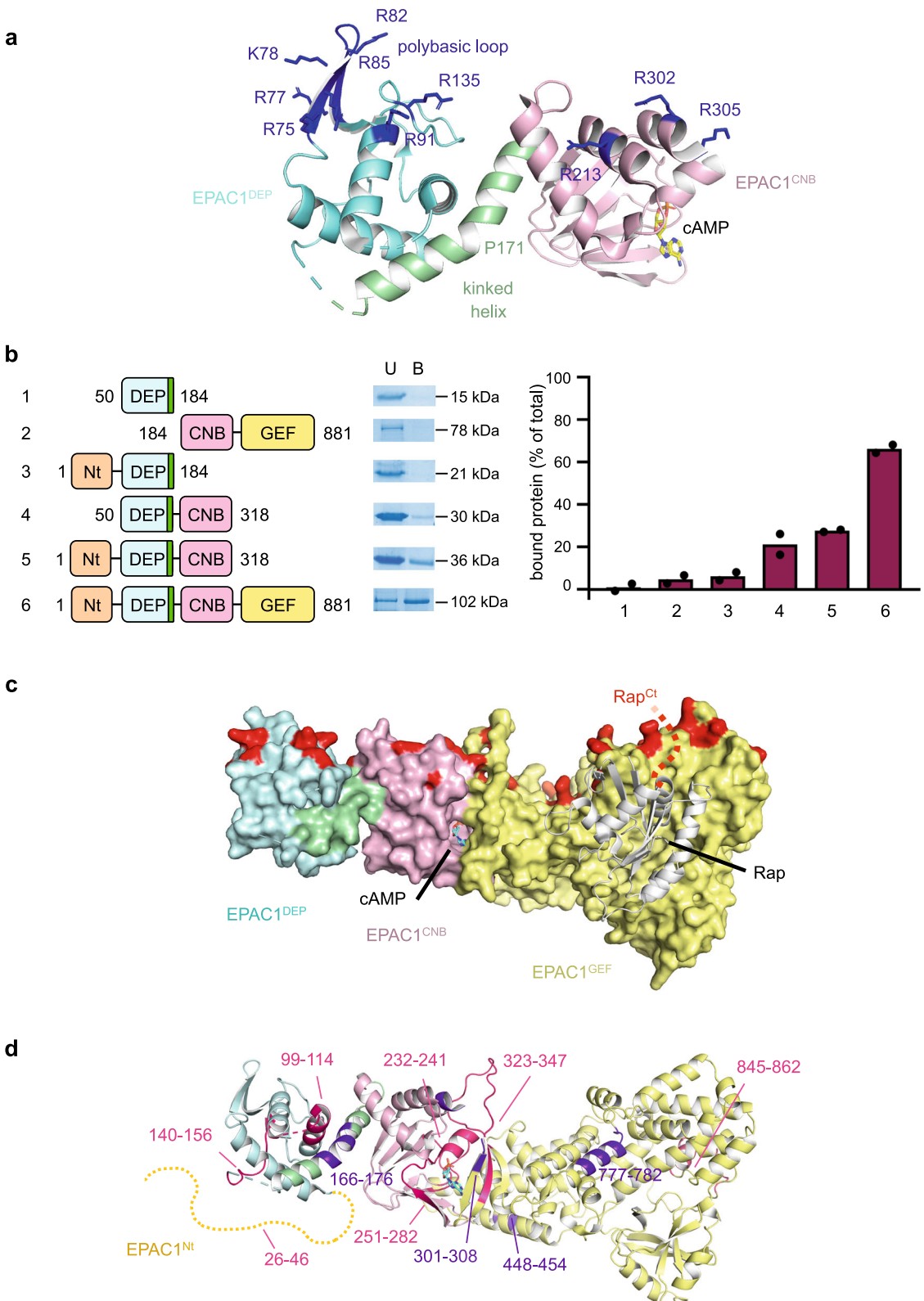

small difference in the relative orientations of the DEP and CNB domains (Supplementary Fig. 2a). This similarity, together with the fact that the two copies of EPAC1[DEP-CNB] in the crystallographic asymmetric unit are identical, suggests that the DEP and CNB domains are rigidly associated and behave as a single unit in the course of EPAC activation. Accordingly, the kinked helix, which forms autoinhibitory interactions with the GEF domain (see below, Supplementary Fig. 4d), is predicted to be displaced upon activation as part of the rigid DEP-CNB unit. The structure also shows that cAMP binds to the CNB domain of EPAC1[DEP-CNB] in the same manner as in the EPAC2[CNB-GEF]-Rap1 complex[20], suggesting that interlocking of the CNB and GEF domains by cAMP does not require further rearrangement of the CNB domain. We note a single difference between EPAC1 and EPAC2, located at Gln 270 in EPAC1, which forms a hydrogen bond with the sugar hydroxyl of cAMP

**Fig. 3 | Structural determinants of EPAC1 binding to membranes. a** Crystal structure of EPAC1$^{DEP-CNB}$. The DEP domain is in light blue with the polybasic loop in dark blue, the connecting α-helix in green, and the CNB domain in pink. cAMP is shown in the sticks. Positively charged residues located in the polybasic loop plane are indicated. The same color coding for EPAC1 domains is used in all structural and schematic representations. **b** Binding of EPAC1 to membranes requires all domains. Binding to PM liposomes was measured for the EPAC1 constructs shown on the left panel, using liposome flotation. The middle panel shows representative Coomassie blue staining of bound (B) and unbound (U) protein. The right panel shows the percentage of bound protein as mean values of $n = 2$ independent experiments. **c** Composite model of activated EPAC1$^{DEP-CNB-GEF}$ bound to Rap1. The GEF domain is in yellow. Rap1 is shown in white, with the predicted position of its lipidated polybasic C-terminus represented by a red dotted line. Arg and Lys residues located in the same plane as the positively charged tract of the DEP and CNB domains are shown in red. **d** HDX-MS analysis of the activation of EPAC1 by cAMP and PM liposomes (HDX of EPAC1/cAMP/liposomes - HDX of EPAC1 in solution). Peptides with increased (in violet) or decreased (in red) H/D exchange are indicated. The N-terminal domain, for which no reliable model is available, is indicated by an orange dotted line. Source data are provided as a Source Data file.

that is not seen with the equivalent Lys 405 in EPAC2 (Supplementary Fig. 2b). The potential to form hydrogen bonds at this residue likely contributes to the distinct selectivities of EPAC1 and EPAC2 for cAMP-derived agonists[28].

The DEP domain does not bind to the plasma membrane on its own[36]. The structure confirms that it lacks a positively charged pocket that could accommodate PS or PIP$_2$ headgroups, such as the phosphoinositide-binding pocket of PH domains. Alternatively, EPAC1 could use more than just the DEP domain to interact with membranes, possibly through positively charged residues distributed throughout its surface. To address this question, we compared the binding of various purified EPAC1 constructs (EPAC1$^{DEP}$, EPAC1$^{DEP-CNB}$, EPAC1$^{Nt-DEP}$, EPAC1$^{CNB-GEF}$, see Supplementary Fig. 1a, b) to PM liposomes (Fig. 3b and Supplementary Fig. 2c). EPAC1$^{DEP}$ did not bind to liposomes on its own, consistent with previous cellular assays[36]. Importantly, none of the truncated constructs bound as strongly as EPAC1$^{FL}$, suggesting that all domains contribute to optimal EPAC1 binding to anionic membranes. Combining our EPAC1$^{DEP-CNB}$ structure to a model of EPAC1$^{CNB-GEF}$/Rap1, we then built a composite model of cAMP-activated EPAC1$^{DEP-CNB-GEF}$ bound to Rap1 and used it to examine how EPAC1 binds to anionic membranes. We identified an extended cationic tract at the surface in the EPAC1$^{DEP-CNB-GEF}$ model, comprised of the polybasic loop in the DEP domain (residues $^{75}$RDRKYHLRLYRQ$^{86}$), and numerous lysines and arginines in the CNB and GEF domains (Fig. 3c). Accordingly, docking of the EPAC1 model onto a membrane using the OPM/PPM server[52] predicted that EPAC1 uses this entire surface to bind to the membrane (Supplementary Fig. 2d). This suggests that EPAC1 uses an extended cationic surface to bind to anionic membranes through multiple non-specific electrostatic interactions contributed by both its regulatory and catalytic regions. In this model, EPAC1 can accommodate Rap1 in a position where its polybasic, lipidated C-terminus points towards the membrane, indicating that this orientation is competent for Rap1 activation on the membrane.

To gain further insight into the organization of EPAC1 on membranes, we used HDX-MS, which allows us to simultaneously map conformational changes associated with activation and interactions with membranes (ref. 53, reviewed in refs. 54, 55), to compare inactive EPAC1 in solution to fully active cAMP- and PM liposome-bound EPAC1 (Fig. 3d and Supplementary Fig. 3). Most protected and deprotected peptides are consistent with the model of active EPAC1 bound to the membrane. The strong interlocking of the CNB and GEF domains by cAMP is readily observed by the protection of the cAMP-binding site (peptide 251–282 in the CNB domain and peptide 323–347 in the GEF domain), and by deprotection of the hinge between the CNB and GEF domains (peptide 301–308), which becomes more flexible following activation. Likewise, the deprotection of several peptides is consistent with the release of autoinhibitory interactions. This is notably the case for the conserved kinked helix (peptide 166–176) between the DEP and CNB domains, a major autoinhibitory element (see the spontaneous activity of EPAC1$^{CNB-GEF}$ lacking this helix in Supplementary Fig. 4d), and for peptide 777–782 in the GEF domain, the equivalent of which faces this helix in the structure of autoinhibited EPAC2. Finally, three peptides are protected within the predicted membrane-interacting surface: peptides 99–114 in the DEP domain, peptides 232–241 in the CNB

domain, and peptides covering the region from 845 to 862 in the GEF domain. In addition, the protection of peptides 323–347 in the GEF domain, which projects a long loop within the predicted membrane-facing region, may also reflect protein-membrane interactions. Besides, HDX-MS reveals marked protection of residues 26–46 in EPAC1$^{Nt}$, together with nearby residues 140–156, which are located in the DEP domain opposite to the predicted membrane-binding region. This suggests that EPAC1$^{Nt}$ folds back onto the DEP domain in cAMP-, membrane-activated EPAC1. Overall, HDX-MS thus supports our model of fully active EPAC1 bound to the membrane and reveals that the N-terminal domain forms previously unknown intramolecular interactions in this active conformation.

## Diversion of EPAC1 dynamics by the chemical inhibitor CE3F4

The above analysis indicates that EPAC1 visits multiple structural intermediates during the course of its dual activation by membranes and cAMP, which can be distinguished by their different GEF efficiencies and affinities for cAMP. Chemical inhibitors that divert specific intermediates can inform on their nature. An appealing compound in that regard is the allosteric inhibitor CE3F4, which inhibits cellular EPAC1 functions and Rap1 activation and the in vitro activation of Rap1 by EPAC1$^{α-CNB-GEF}$, a truncated construct that contains the kinked α-helix and the CNB and GEF domains[44]. CE3F4 does not compete with cAMP[44], and it also does not compete with the Rap1 GTPase directly, as shown by the fact that it does not inhibit the isolated GEF domain (Supplementary Fig. 4a, ref. 44). Using purified proteins, we confirmed that CE3F4 inhibits the GEF activity of cAMP-activated full-length EPAC1 in solution (Fig. 4a and Supplementary Fig. 4b).

To determine whether CE3F4 disrupts the conformational landscape of EPAC1, we analyzed its effect on the structures of auto-inhibited and cAMP-activated EPAC1$^{α-CNB-GEF}$ in solution, using synchrotron SAXS coupled to size exclusion chromatography (SEC-SAXS). SEC-SAXS informs on the ensemble of conformations in a given state, including the largest protein dimension ($D_{max}$). Data acquisition and analyses are summarized in Supplementary Table 2 and Supplementary Fig. 5. The conformational change between EPAC1$^{α-CNB-GEF}$ and cAMP-EPAC1$^{α-CNB-GEF}$ was readily detected as an increase in $D_{max}$ from 107 to 130 Å. CE3F4 decreased the $D_{max}$ of cAMP-EPAC1$^{α-CNB-GEF}$ to 114 Å (Fig. 4b), indicating that it induces an alternative conformational ensemble that is distinct from either autoinhibited or cAMP-activated EPAC1$^{α-CNB-GEF}$.

To get further insight into how CE3F4 disrupts the conformational landscape of EPAC1, we compared how it affects EPAC1 properties in solution and in the presence of membranes. First, we asked whether CE3F4 affects the interaction of EPAC1 with the small GTPase Rap1, using size exclusion chromatography (Fig. 4c and Supplementary Fig. 4c). Formation of the Rap1-EPAC1$^{α-CNB-GEF}$ complex was readily observed in the presence of cAMP. In striking contrast, CE3F4 impaired this interaction. Since CE3F4 does not inhibit the GEF domain directly (Supplementary Fig. 4a), this implies that it induces inhibitory rearrangements in the Rap1-binding site of the GEF domain in an allosteric manner. Next, we asked whether cAMP is necessary for this allosteric mechanism. For that, we assayed the effect of CE3F4 under conditions where partial activity is observed in the absence of cAMP. In solution,

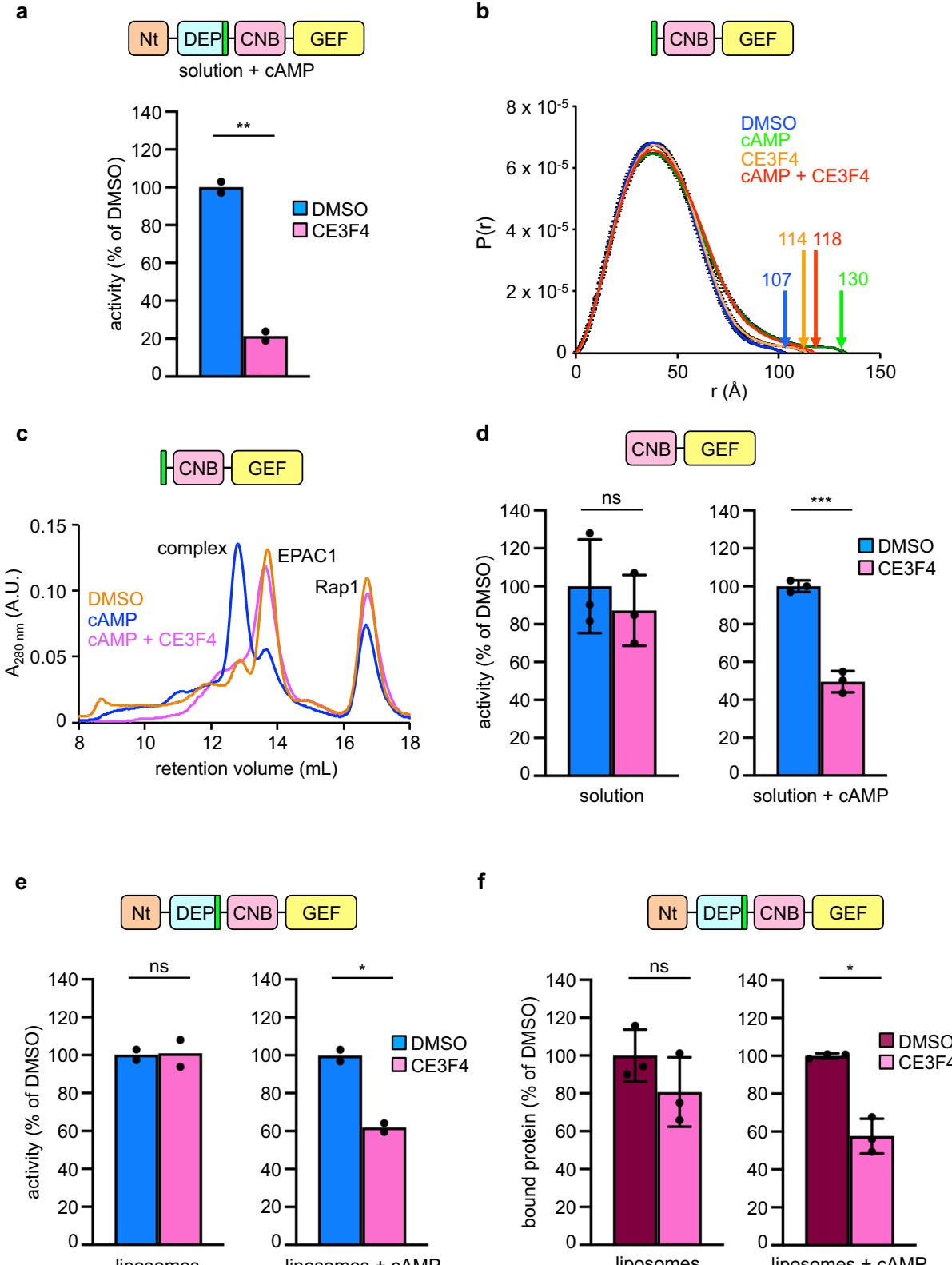

we used EPAC1^CNB-GEF, a construct that lacks the helix located between the DEP and CNB domains which is predicted to form autoinhibitory interactions with the GEF domain in the absence of cAMP. EPAC1^CNB-GEF displays intrinsic GEF activity in the absence of cAMP, confirming that removal of the helix results in a partial release of autoinhibition, and this intrinsic activity is increased by cAMP (Supplementary Fig. 4d[22]). CE3F4 did not inhibit EPAC1^CNB-GEF in the absence of cAMP, but

inhibition was recovered in the presence of cAMP (Fig. 4d). We then took advantage of the fact that EPAC1^FL is partially active on membranes in the absence of cAMP to investigate whether CE3F4 also requires cAMP in the context of the membrane. CE3F4 did not inhibit liposome-activated EPAC1^FL in the absence of cAMP, while inhibition was recovered upon the addition of cAMP (Fig. 4e and Supplementary Fig. 4e). These data indicate that CE3F4 inhibits EPAC1 only in the

**Fig. 4 | The CE3F4 inhibitor diverts the conformational landscape of EPAC1.**
**a** EPAC1$^{FL}$ is inhibited by CE3F4 in solution. Nucleotide exchange kinetics were measured with Rap1 at 500 nM, EPAC1$^{FL}$ at 100 nM, cAMP at 300 μM and R-CE3F4 at 50 μM. **b** CE3F4 diverts the conformational landscape of EPAC1$^{α\text{-}CNB\text{-}GEF}$. Unbound and cAMP-bound EPAC1$^{CNB\text{-}GEF}$ were analyzed by synchrotron SEC-SAXS in the absence or presence of CE3F4 (100 μM, racemic mix). The distance distribution functions P(r) are plotted at the same scale. The $D_{max}$ values (Å) are indicated. SDs (in black) are estimated by Monte-Carlo simulations using ATSAS. **c** CE3F4 impairs the formation of the Rap1/EPAC1$^{α\text{-}CNB\text{-}GEF}$ complex. Complex formation was analyzed by size exclusion chromatography in the presence of EDTA with DMSO alone (orange), cAMP (blue) or cAMP + CE3F4 (pink). AU absorbance units. **d** CE3F4 requires cAMP to inhibit constitutively active EPAC1$^{CNB\text{-}GEF}$ in solution. Nucleotide exchange kinetics experiments were performed as in **a**. **e** CE3F4 requires cAMP to

inhibit liposome-activated EPAC1$^{FL}$. Nucleotide exchange kinetics experiments were carried out as in **a**, with 100 μM PM liposomes. **f** CE3F4 reduces the binding of EPAC1 to liposomes in the presence of cAMP. Binding was analyzed by liposome flotation using PM liposomes. cAMP is at 1 mM, CE3F4 at 50 μM. The EPAC1 construct used in each experiment is shown at the top of the panel. Data were presented as means of $n = 2$ independent experiments (**a**, **e**), or means ± SD of $n = 3$ independent experiments (**d**, **f**). The mean value obtained with DMSO was set to 100% in panels **a**, **d**, **e**, **f**. Statistical analysis was performed by two-tailed unpaired Student's $t$-tests. Welch's correction was applied in **a**, **e**, **f** (right). ns: $P > 0.05$, not significant; *$P < 0.05$; **$P < 0.01$; ***$P < 0.001$. Test statistics, mean values with 95% CI, effect sizes, and exact $P$ are in Supplementary Table 5. Source data are provided as a Source Data file.

presence of cAMP, whether in solution or on the membrane. Finally, we analyzed whether the allosteric structural changes induced by CE3F4 interfere with the interaction of EPAC1$^{FL}$ with membranes. Consistently, CE3F4 reduced the binding of EPAC1$^{FL}$ to PM liposomes only in the presence of cAMP (Fig. 4f), suggesting that the structural elements that block the binding of Rap1 are located close to the membrane interface.

Together, our findings indicate that CE3F4 specifically recognizes cAMP-activated EPAC1 intermediates, and that it acts by remodeling structural elements that respond to cAMP in a manner that blocks access to the Rap-binding site in the vicinity of the membrane.

## Discussion

Previous studies identified membranes as important determinants of EPAC1-Rap1 signaling by cAMP in cells[36,37], yet how membranes shape the behavior of EPAC1 has remained poorly understood. Here, we discovered that anionic membranes are pivotal regulators of the response of EPAC1 to cAMP by increasing the affinity of EPAC1 to cAMP by two orders of magnitude. Our data reveal that EPAC1 has a complex structural and dynamic landscape, which culminates with a highly active form activated by both cAMP and membranes acting in synergy. Remarkably, the different states can be distinguished by the EPAC1 inhibitor CE3F4, which uncovers important aspects of the inhibitory mechanism. We discuss below the implications of these findings for EPAC1 regulation by membranes, for cAMP signaling in cells, and for drug discovery.

Membranes are increasingly recognized as pivotal determinants of small GTPase functions (reviewed in refs. 30, 31), notably by regulating the activity of their regulators (e.g., refs. 56, 57). To understand the inner workings of small GTPase signaling pathways of biomedical interest, it is thus essential to analyze and quantify the contribution of membranes. Our study provides important insights into this aspect of EPAC1 functions. Our results demonstrate that EPAC1 has at least four states in equilibrium (Fig. 5a): a soluble apo form which is fully auto-inhibited, two modestly active forms bound to either cAMP or membranes, and an optimally active form that is both cAMP- and membrane-bound. We propose that membranes relieve autoinhibition by repositioning the regulatory domains away from the catalytic site, and that this open conformation remains highly dynamic until it is interlocked by the binding of cAMP to yield full activation. In addition to this direct activating effect, localization of EPAC1 and Rap GTPases at the membrane surface may increase their encounter probability through dimensional reduction, which may add to the GEF efficiency of EPAC1 on membranes. Our study further suggests that EPAC1 is recruited to the plasma membrane by multiple non-specific electrostatic interactions with PS and phosphoinositide lipids, which are major constituents of this membrane (reviewed in ref. 58). These protein/membrane interactions likely compensate for the loss of autoinhibitory intramolecular interactions of cytosolic EPAC1, providing an energetic rationale for the ability of the membrane to act as a moderate activator of EPAC1 on its own. Given the potent effect

measured for liposomes containing PS and PIP$_2$ lipids, it is surprising that the PA-specific recruitment of EPAC1 to the plasma membrane[37] was not recapitulated by PA-containing liposomes. Since PA is an essential precursor in the synthesis of major plasma membrane phospholipids, including PS and phosphoinositides (reviewed in ref. 59), a tentative explanation is that the removal of PA in cells impairs EPAC1 recruitment indirectly by affecting the content of the plasma membrane in these lipids.

The orientation of GEFs with respect to the membrane has been shown to be a critical determinant of their activity[60,61]. Here, the predicted electrostatic interface of EPAC1 with the membrane suggests that EPAC1 is closely apposed to the membrane and precisely oriented to activate membrane-associated Rap1. We propose that this allows electrostatic coincidence detection of EPAC1 and Rap1 at the plasma membrane through their concomitant interactions with anionic lipids (Fig. 5b). Such a mechanism may, in turn, be important for the specificity of EPAC1 for Rap GTPase isoforms. Like Rap1A, the Rap1B isoform carries numerous positively charged residues in its C-terminus (RKTPVPGKARKKSSCQLL). In contrast, the related Rap2 isoforms have only one or no such residue (Rap2A: YAAQPDKDDPCCSACNIQ, Rap2B: YAAQPNGDEGCCSACVIL, and Rap2C: YSSL-PEKQDQCCTTCVVQ). Thus, Rap1 isoforms have determinants for electrostatic coincidence detection with EPAC1 at the plasma membrane, while Rap2 isoforms lack such determinants. Alternatively, Rap2 isoforms may be activated in other locations where EPAC1 has been observed, such as the mitochondria or the nuclear pore (reviewed in ref. 62), or solely by RapGEF subfamilies other than EPAC (reviewed in ref. 63). Because of its shared domain organization, it is likely that the related EPAC2 protein follows a similar membrane-binding and activation scenario. Future studies are needed to identify the specific determinants for alternative subcellular EPAC1 targeting and for the specificity of regulation of EPAC2 by membranes.

Next, our study has implications for cAMP signaling by EPAC1 in cells. The actual concentration of cAMP at any moment and subcellular location depends on a complex balance between synthesis, degradation, buffering, and diffusion processes, which ultimately should match the localization and the cAMP affinity of specific cAMP signal transducers (reviewed in refs. 1, 2, 7). Seminal studies recently established that cAMP is restricted within receptor- and membrane-proximal nanodomains, thus, does not diffuse freely in the cytosol where its concentration remains low[3–6]. By identifying membranes as a previously overlooked component of cAMP signaling through EPAC1, our findings bring important insights into this issue.

Notably, they resolve the conundrum of the intriguingly low affinity of soluble EPAC1 for cAMP, which makes it unlikely that EPAC1 can be activated in the bulk cytosol given the low cAMP concentration and variations in this compartment[6]. Hence, soluble cAMP-bound EPAC1 is likely a negligible species in the cell. In contrast with soluble EPAC1, membrane-attached EPAC1 has a high affinity for cAMP, suggesting that in cells, EPAC1 must be membrane-attached to be activated by cAMP. We propose that membranes prime EPAC1 for subsequent

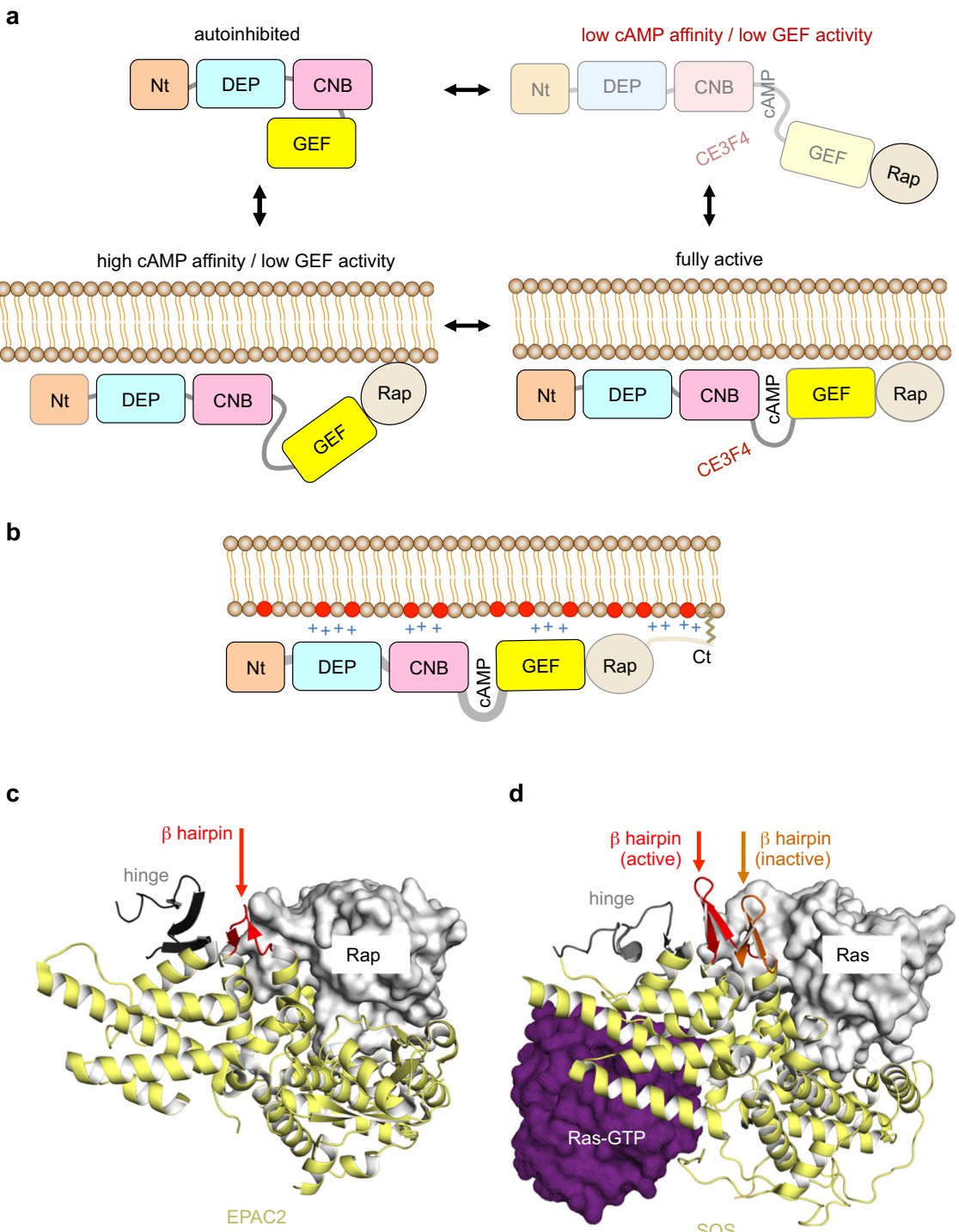

**Fig. 5 | Models of EPAC1 activation by cAMP and membranes and inhibition by CE3F4. a** The conformational ensemble of EPAC1. EPAC1 exists in at least four states in equilibrium (arrows) with distinct GEF activities and cAMP affinities. Because of its low affinity for cAMP in solution, the soluble cAMP-bound EPAC1 intermediate (light colors, top right) is predicted to be a negligible species in cells. For the same reason, it is also predicted that in cells, CE3F4 targets only membrane-associated, cAMP-activated EPAC1 (bottom right). **b** Electrostatic coincidence detection of EPAC1 and Rap1A on anionic membranes. Positive charges in proteins (blue crosses), negative charges in the membrane (red circles), and the lipidated C-terminus of the Rap GTPase are indicated. **c** β-hairpin-mediated allosteric

inhibition of EPAC1 by CE3F4. The binding of CE3F4 to the hinge located between the CNB and GEF domains (in black) is proposed to displace a conserved β-hairpin (in red) into the Rap-binding site. The hinge and the β-hairpin structures are from the crystal structure of cAMP-activated EPAC2[20] (from PDB entry: 3CF6). **d** An equivalent inhibitory β hairpin in the related RasGEF SOS. In inactive SOS, the β-hairpin protrudes into the Ras-binding site (orange). In Ras-GTP-activated SOS, it has relocated outside the Ras-binding site (in red). The relative positions of the β-hairpin are based on the superimposition of the Ras-binding site in inactive SOS and Ras-GTP-activated SOS structures (ref. 65, from PDB entries: 1NXV and 2II0).

activation by a cAMP rise. Accordingly, activation of EPAC1 by cAMP likely takes place almost exclusively at the membrane, with cAMP securing the localization of EPAC1 at the membrane rather than driving it. In this scheme, an alternation between soluble and membrane-attached EPAC1 is likely spontaneous, with EPAC1 falling off the membrane if it does not encounter a cAMP signal.

Recent studies showed that nanometer-sized cAMP compartments associated with GPCRs translate into localized PKA activation to ensure receptor-specific signaling[6]. Specific activation of EPAC1 by different β-adrenergic receptors has been reported, which resulted in the stimulation of distinct signaling arms in cardiac myocytes[64], raising the question of whether EPAC1 may also respond to specific receptors through cAMP nanodomains. Our data reveal that EPAC1 has biochemical determinants to function in membrane-proximal cAMP nanodomains, which may explain the mechanism of such receptor-specific responses (Supplementary Fig. 6). Notably, while soluble EPAC1 is not activated at the low cAMP concentration encountered in the cytosol, membrane-attached EPAC1 is highly sensitive to cAMP concentrations in the range of those measured in receptor-associated nanodomains. In addition, our data resolve the intriguing discrepancy between the sensitivities of PKA and EPAC1 to cAMP fluctuations by showing that upon membrane binding, the response of EPAC1 to cAMP (affinity and concentration resulting in half-maximal activation) is similar to that of PKA[24,25]. Thus, cAMP nanodomains may control both PKA and EPAC pathways.

Together, our study revisits the molecular mechanisms that enable EPAC1 to transduce cAMP signaling in cells. Important questions for future studies will be to document how membrane-restricted activation of EPAC1 articulates with compartmentalized cAMP.

Finally, our findings have implications for drug discovery. The growing evidence that EPAC proteins play pivotal roles in physiological and pathogenic conditions has motivated intense research to discover specific agonists and antagonists (reviewed in ref. 43). Our findings provide a framework for their discovery and improvement.

First, our study identifies previously unknown conformations of EPAC1, each featuring distinct intramolecular, protein/membrane, and protein/ligand interactions (Fig. 5a). Each of these states thus offers unique structural characteristics that can be exploited to discover activating and inhibitory drugs. Importantly, soluble cAMP-activated EPAC1 appears an irrelevant target for inhibition, as this intermediate is likely in negligible amounts in cells. Alternatively, partially activated, membrane-bound EPAC1 may constitute an interesting target for chemicals that bind at the EPAC1-membrane interface, as recently shown by a proof-of-concept GEF inhibitor, Bragsin, which acts by binding at the interface between an ArfGEF and the membrane to alter its orientation[61].

Second, our findings provide insights into the cellular mechanism of action of agonists that compete with cAMP. Notably, we find that the affinity of cAMP for membrane-bound EPAC1 is only slightly higher than that of the widely used super agonist 007 for soluble EPAC1[25–28]. Thus, 007 may bypass the requirement that EPAC1 is bound to the membrane to be activated. Accordingly, it may yield non-physiological activation of EPAC1 in the cytosol and/or on alternative membranes, hence miswire EPAC1 signaling. Such side effects should be considered when using this and other cAMP analogs to interrogate the biology of EPAC proteins in the cell.

Finally, our results provide insight into the molecular and cellular mechanism of action of the cell-active CE3F4 inhibitor. First, the fact that the binding site for CE3F4 exists only in cAMP-bound EPAC1 intermediates, of which soluble cAMP-activated EPAC1 is a negligible species at physiological cAMP concentrations, makes it likely that, in cells, the actual target of CE3F4 is membrane- and cAMP-bound EPAC1 (Fig. 5a). Second, our data suggest that CE3F4 binds to a cAMP-responsive region outside the GEF domain in a manner that projects an inhibitory element into the Rap1-binding site. We propose that this cAMP-responsive region is the hinge between the CNB and GEF

domains, and that the inhibitory element is a flexible β-hairpin of the GEF domain located between the hinge and the Rap1-binding site (EPAC2: [919]GNKTFIDNLVN[929], EPAC1: [786]GNHTLVENLIN[796]; Fig. 5c). In SOS, a Ras GTPase exchange factor whose GEF domain is related to that of EPAC, the equivalent β-hairpin ([943]GNPEVLKRHGKELIN[957]) hinders access to the Ras-binding site, from which it is removed upon allosteric activation of SOS by Ras-GTP[65] (Fig. 5d). We propose that CE3F4 acts in a similar manner by projecting the β-hairpin into the Rap-binding site, using the flexible Gly/Asn residues at both ends of the β-hairpin to support this movement. Together, our study reveals that CE3F4 has a remarkable allosteric mechanism which requires that EPAC1 is fully activated to be inhibited in cells. By targeting EPAC1 "in action", such a mechanism makes CE3F4 highly specific to a membrane-localized cAMP-EPAC1-Rap1 signaling axis in cardiovascular conditions and other diseases.

In conclusion, our findings provide a robust framework to understand the molecular basis of EPAC1 activation by cAMP at the surface of membranes in cells and in diseases, and the action of activators and inhibitors. Future studies are now needed to establish how EPAC1 and EPAC2 respond to compartmentalized cAMP signaling within this framework, and for mechanism-based improvement of the CE3F4 inhibitor. More generally, our work highlights the importance of designing drug discovery strategies that take full advantage of elusive Achille's heels, such as structural malleability or protein-membrane interactions, which are critical in regulating small GTPase signaling in physiological functions and in diseases.

## Methods

### Protein cloning, expression, and purification

Primer sequences are in Supplementary Table 3. Human EPAC1[DEP] (residues 50–184), EPAC1[Nt-DEP] (residues 1–184), EPAC1[DEP-CNB] (residues 50–318), EPAC1[Nt-DEP-CNB] (residues 1–318), EPAC1[CNB-GEF] (residues 184–881) and EPAC1[α-CNB-GEF] (residues 149–881) were cloned into the Gateway destination vector pHMGWA carrying a 6xHis-MBP tag in N-terminus. Human EPAC1[GEF] (residues 321–881) was cloned into the Gateway destination vector pETG20a carrying a 6xHis-TRX tag in N-terminus. Human full-length EPAC1 (EPAC1[FL]) was cloned into a pET28a vector (6xHis tag in N-terminus). A TEV protease cleavage site was inserted during cloning after the 6xHis-MBP, 6xHis-TRX, or 6xHis N-terminal tags of each vector. Full-length human Rap1A was cloned into the pET3a vector carrying a 6xHis tag in C-terminus. All clones were confirmed by sequencing (GATC Biotech). All EPAC1 constructs and Rap1 were produced in the E. coli BL21 (DE3) pG-KJE8 chaperone-expressing strain[66], in LB medium with 0.5 g/L L-arabinose and 2.5 mg/L tetracycline. Induction of proteins was done with 0.5 mM IPTG overnight at 20 °C. Bacterial pellets were resuspended in lysis buffer (50 mM Tris-HCl pH 8.0, 300 mM NaCl, 2 mM β-mercaptoethanol, 2 mM MgCl2, 0.2% n-Dodecyl β-D-maltoside, 0.5 mg/mL lysozyme, 7.5 U/mL benzonase, and anti-protease cocktail) and cells were broken using a French press. After centrifugation at 15,000×g for 30 min, the cleared lysate supernatant was filtered over a 0.22 μm filter and loaded onto a Ni-NTA affinity chromatography column (HisTrap FF, GE Healthcare) and His-tagged proteins were eluted with 250 mM imidazole. For all proteins except EPAC1[FL] and Rap1, the 6xHis tag was cleaved by the TEV protease (1/10 w/w ratio) during overnight dialysis in storage buffer (20 mM Tris-HCl pH 8.0, 150 mM NaCl, 2 mM β-mercaptoethanol, 2 mM MgCl2, 2.5% glycerol, 0.02% DDM) at 4 °C. The cleaved tag was removed by a second Ni-NTA affinity chromatography step. The purity of all proteins was polished on either a Superdex 75 or a Superdex 200 16/600 size exclusion column equilibrated with storage buffer. Protein purity was assessed by SDS-PAGE.

### Liposome preparation and binding assay

Lipids (natural origin) were from Avanti Polar Lipids (catalog numbers: PS: 840032; PE: 840022; PC:840054; PIP2: 840046; fatty acid

distributions as indicated by the supplier), except NBD-PE (Sigma). Liposomes were prepared as described in ref. [61] in a buffer containing 50 mM HEPES pH 7.4 and 120 mM potassium acetate. All liposomes contain phosphatidylethanolamine (PE) at 20% and other lipids as indicated, and are completed to 100% by phosphatidylcholine (PC). PM liposomes contain 48% PC, 20% PE, 10% phosphatidylserine (PS), 20% cholesterol, and 2% phosphatidylinositol-4,5-bisphosphate (PIP$_2$). All liposomes were extruded through a 0.2-μm-pore size filter before use, and size distributions were checked by dynamic light scattering (DLS) on a DynaPro instrument (Wyatt). The binding of proteins to liposomes was assayed by liposome flotation as described in ref. [57]. When indicated, 1 mM cAMP was added to the sucrose gradient. 1/8 of the bottom and top fractions were analyzed by SDS-PAGE using Coomassie blue staining. Quantification of the intensity of the bands was done using Image Lab™ v6.0 on a ChemiDoc MP Imaging System (Biorad).

### Nucleotide exchange kinetics assay

For kinetics assays, the Rap1 GTPase was loaded with BODIPY-FL-GDP (Jena Bioscience) prior to nucleotide exchange by incubating 50 μM Rap1 with 250 μM BODIPY-GDP and 10 mM EDTA for 30 min at room temperature. Nucleotide exchange was stopped by the addition of 75 mM MgCl$_2$. Removal of excess nucleotides and buffer exchange were done on a PD SpinTrap G-25 (GE HealthCare Life Science). Nucleotide exchange kinetics were monitored by following the decay of BODIPY-GDP fluorescence upon replacement by GTP, using excitation and emission wavelengths of 485 and 530 nm, respectively. Measurements were carried out at 30 °C on a Cary Eclipse fluorimeter (Varian) under stirring in 800 μL cuvettes, or in microplate wells (150 μL) on a FLEXstation (Molecular Devices). In all experiments, proteins and liposomes were preincubated in HKM buffer (HEPES 50 mM pH 7.4, 120 mM potassium acetate, 1 mM MgCl$_2$) for 1 min under stirring before the exchange reaction was initiated by the addition of 100 μM GTP. cAMP was added as indicated. Liposomes were added at 100 μM (total lipids). For kinetics in solution, $k_{obs}$ values were determined from a mono-exponential fit. We noted that kinetics in the presence of membranes were best fit by one fast and one slow exponentials. These two components likely correspond to the activation rates of membrane-associated and soluble Rap1, respectively. In this case, $k_{obs}$ values corresponding to the fast component were used.

### Catalytic efficiency, dissociation constant, and maximal activity

To determine EPAC1 catalytic efficiency in solution with cAMP, or on liposomes with or without cAMP, nucleotide exchange reactions were performed at different concentrations of EPAC1. $k_{obs}$ values were plotted as a function of the concentration of EPAC1. Nonlinear regression based on a straight line model was performed with GraphPad Prism. EPAC1 pseudo first order catalytic efficiencies $k_{cat}/K_M$ were deduced from the slopes of the curves, according to the Michaelis–Menten formalism[67]. To determine the dissociation constants ($K_D$) and maximal activities ($k_{max}$) in the presence of cAMP or 007, in solution or on liposomes, nucleotide exchange reactions were performed in the presence of different concentrations of cAMP or 007. $k_{obs}$ values were graphed in semi-log plots as a function of [cAMP] or [007]. In order to plot $k_{obs}$ measured in the absence of agonist, the zero concentration was approximated to $10^{-11}$ nM, at which less than one molecule of cAMP or 007 is expected to be present in the reactions. Nonlinear regression based on a sigmoidal model was performed with GraphPad Prism. $K_D$ and $k_{max}$ were deduced from the EC50 and top values of the curves, respectively.

### Crystallographic analysis

Crystallization screens were performed using the sitting-drop vapor diffusion method at 18 °C with a Mosquito robot (TTP LabTech) in 96-well crystallization plates by mixing 100 nL of EPAC1$^{DEP-CNB}$-cAMP (5 mg/mL EPAC1, 10 mM cAMP) with 100 nL of precipitant solution.

Crystals were obtained in 20% w/v polyethylene glycol 3350, 100 mM BIS-TRIS propane pH 6.5, 100 mM sodium phosphate, 100 mM potassium phosphate. Crystals were cryo-protected using the reservoir solution supplemented with 15% glycerol prior to flash freezing. A complete diffraction dataset was collected on ID30B beamline (ESRF synchrotron, France) and was integrated with the program AutoProc[68]. The structure was solved with the automated molecular replacement pipeline BALBES (CCP4 suite)[69] using the CNB domain of activated EPAC2 as a model (from PDB entry 3CF6[20]). Refinement was carried out with the program Phenix[70] using TLS parameters, in alternation with graphical building using Coot41[71]. The quality of the structure was assessed using MolProbity42. Data collection and refinement statistics are reported in Supplementary Table 1.

### Structural modeling

The GEF domain of EPAC1 was modeled using the protein fold recognition Phyre2 server[72]. The composite model of activated EPAC1$^{DEP-CNB-GEF}$ bound to Rap1 was generated by superimposing our EPAC1$^{DEP-CNB}$-cAMP crystal structure and the model of EPAC1$^{GEF}$ onto the crystal structure of EPAC2$^{CNB-GEF}$-Rap1-cAMP[20], (PDB entry 3CF6). Docking of the EPAC1 model onto an anionic lipid membrane was done with the OMP server using default parameters[52].

### HDX-MS

EPAC1$^{FL}$ (15 μM) was deuterated at room temperature alone or in a complex with cAMP (4.5 mM) and freshly prepared PM liposomes (3.125 mM total lipid). To optimize protein stability over time, a new mixture was freshly prepared every three replicates. Deuteration was carried out by incubating 3 μL of the EPAC1 solutions into 57 μL of deuterium buffer (50 mM HEPES pD 8.0, 150 mM NaCl, 2 mM β-mercaptoethanol, 2 mM MgCl$_2$, and D$_2$O) for defined durations (0.5, 1, 2, 5, and 10 min), corresponding to a 95% deuterated final solution. All conditions were analyzed in triplicate. About 55 μL of each deuterated preparation were quenched with 55 μL of quench buffer (150 mM Glycine pH 1.5 and 4 M guanidine-HCl) at 1 °C for 30 s with a final pH of 2.4. Deuteration, quenching, and injection of the samples were automatically performed, three by three, using a Leap HDX Automation manager (Waters). Digestion and chromatography were done on an Acquity UPLC system with HDX technology (Waters, Manchester, UK). Digestion was performed on a pepsin-immobilized cartridge (Enzymate pepsin column, 300 Å, 5 μm, 2.1 mm I.D. × 30 mm, Waters, Manchester, UK) at a 100 μL/min flow rate of 0.1% formic acid solution at 20 °C. Peptides were then trapped on a UPLC pre-column (ACQUITY UPLC BEH C18 Van-Guard pre-column, 2.1 mm I.D. × 5 mm, 1.7 μm particle diameter, Waters) and separated on a UPLC column (ACQUITY UPLC BEH C18, 1.0 mm I.D. × 100 mm, 1.7 μm particle diameter, Waters) at 0.1 °C. Peptide separation was performed at a flow rate of 40 μL/min, with an elution gradient of solvent A (0.1% formic acid, water) and solvent B (0.1% formic acid, acetonitrile) from 2–40% B over 7 min followed by a 0.5 min ramp to 85% B. Peptides were then analyzed with a Synapt G2Si HDMS (Waters, Wilmslow, UK) with Glu-fibrino peptide infused for calibration and lock mass correction in positive ion and resolution mode. MS/MS acquisition were done using data-independent acquisition (MS$^E$) on a range of 50–2000 m/z. Protein Lyse Global Server 2.5.3 (PLGS, Waters) searches were done to identify unlabeled peptides using a homemade database with the protein of interest and pepsin sequences with methionine oxidation as variable modification sites. DynamX 3.0 (Waters) was next employed to filter identified peptides with a minimum intensity of 1000, a minimum fragment of 0.3 per amino acid and their presence in at least two out of three replicates. Isotopic profiles for all identified peptides were manually checked and validated without back-exchange correction. Deuteration mean values were reported as relative. Differences in deuterium uptake were statistically validated with a $p$ value of 0.01 with the MEMHDX software[73] with statistical significance thresholds

set to 0.01. HDX results were mapped onto the model of fully active EPAC1[FL]. HDX-MS data were provided as a Source Data File and have been deposited in the PRIDE repository[74].

## Inhibition experiments
All inhibition experiments were done with either the active R-CE3F4 enantiomer or with the CE3F4 racemic mix, which contains >90% of the active enantiomer, synthesized as described in ref. 75. CE3F4 was prepared at 50 mM in DMSO and used at a final concentration of 50 μM except when indicated otherwise. Control experiments without CE3F4 were performed in the presence of vehicle (DMSO).

## SAXS data acquisition and analysis
All small angle X-ray scattering (SAXS) data were obtained from size exclusion chromatography-SAXS (SEC-SAXS) experiments on the SWING beamline (SOLEIL synchrotron, France), using the EPAC1[α-CNB-GEF] construct. EPAC1 was used at a concentration of 60 μM to obtain high SAXS intensities. Samples were injected on high-performance liquid chromatography (HPLC) size exclusion column (BioSEC-3 300 Å, Agilent Technologies, Inc.) equilibrated with the elution buffer (20 mM Tris-HCl pH 8.0, 150 mM NaCl, 0.5 mM EDTA). For SEC-SAXS experiments with CE3F4, EPAC1 was preincubated with 100 μM CE3F4 (racemic mix) and the elution buffer was supplemented with 100 μM CE3F4. For SEC-SAXS experiments with cAMP, EPAC1 was activated with 1 mM cAMP, and the elution buffer was supplemented with 200 μM cAMP. Data reduction to absolute units, frame averaging, and subtraction were done using the FOXTROT program suite (SOLEIL synchrotron). Frames corresponding to the high-intensity fractions of the peak and having a constant radius of gyration ($R_g$) within error were averaged. Data analyses were performed with programs from the ATSAS package36 suite[76] and are summarized in Supplementary Table 2. $R_g$ values were evaluated by Guinier Wizard using the data within the range of Guinier approximation $sR_g < 1.3$ and by Distance Distribution Wizard, both of which are modules of the PRIMUS program. The maximum distance $D_{max}$ was estimated with PRIMUS and refined by trial and error with GNOM. The distance distribution functions were calculated with GNOM. SAXS profiles were compared with DATCMP (ATSAS suite).

## Analysis of complex formation by size exclusion chromatography
All experiments were carried out in elution buffer (20 mM Tris-HCl pH 8.0, 150 mM NaCl, 0.5 mM EDTA) containing DMSO or 100 μM CE3F4 (racemic mix). EPAC1[α-CNB-GEF] was used at 25 μM in all experiments. EPAC1 was preincubated with DMSO or 100 μM CE3F4 for 5 min, before the addition of Rap1 (75 μM), cAMP (1 mM), and EDTA (5 mM). Proteins samples were injected into a 25 mL Superdex 200 gel filtration column (GE Healthcare) at 0.5 mL/min. Fractions were analyzed by SDS-PAGE.

## Statistics
Data are presented as means, and error bars correspond to standard deviation (SD) or standard error (SE), as indicated in the figure legends. All statistical analyses were performed with GraphPad Prism 8.4.3. Two-tailed unpaired Student's $t$-test was used to compare two sets of data. Welch's correction was applied if an $F$-test indicated significantly different variances. Ordinary one-way ANOVA was used to compare more than two sets of data having the same SDs, followed by Sidak's test for multiple comparisons between selected sets of data, or Dunnett's test for multiple comparisons with control. Brown–Forsythe ANOVA was run instead, followed by Dunnett's T3 multiple comparison test, if a Brown–Forsythe or Bartlett's test indicates significantly different SDs. Adjusted $P$ values are given following multiple comparison tests. $P$ value intervals are summarized by asterisks in the figures and explicited in the legends ($\alpha = 0.05$). Test statistics ($t, q, DF$),

mean values with 95% confidence interval (CI), effect sizes, and exact $P$ values are provided in Supplementary Table 4 (multiple comparison tests) and Supplementary Table 5 ($t$-tests).

## Reporting summary
Further information on research design is available in the Nature Portfolio Reporting Summary linked to this article.

## Data availability
Coordinates and structure factors of the X-ray crystallography structure of EPAC1[DEP-CNB]-cAMP have been deposited in the Protein Data Bank under accession code 6H7E. PDB codes of previously published structures used in this study are 3CF6, 1NXV, and 2II0. SAXS data have been deposited in the SASBDB database under accession codes SASDQA8 (unbound EPAC1), SASDQB8 (EPAC1-cAMP), SASDQC8 (EPAC1-CE3F4), and SASDQD8 (EPAC1-cAMP-CE3F4). HDX-MS data have been deposited in the PRIDE repository with accession code PXD040227.

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

## Acknowledgements

This study was supported by grants to J.C. from the Laboratory of Excellence LERMIT supported by the French National Research Agency (ANR-10-LABX-33) under the program "Investissements d'Avenir" ANR-11-IDEX-0003-01, the Fondation pour la Recherche Médicale (grants DEQ20150331694 and EQU202003010344) and the Institut National du Cancer (grant 2014-160). F.P. was supported by a PhD grant from the Ecole Normale Supérieure Paris-Saclay, C.S. by a PhD grant from the LERMIT program. R.F. was supported by a grant from the Leducq Foundation for Cardiovascular Research (grant 19CVD02). The authors acknowledge the support and the use of resources of the French Proteomic Infrastructure ProFI ANR-10-INBS-08-03, the GIS IBiSA, and Région Alsace for financial support in purchasing a Synapt G2SI HDMS instrument. We are grateful to the scientific staff at the SAXS beamline SWING at synchrotron SOLEIL (Gif-sur-Yvette, France) and the X-ray crystallography beamline ID30 at synchrotron ESRF (Grenoble, France) for their expert help and advice. We thank Olivier Duclos and Magali Mathieu (SANOFI) for providing SAXS beamline time at synchrotron SOLEIL, Gérald Peyroche (Cherfils lab, CNRS and ENS Paris-Saclay) for his help and advice throughout the study and Ilham Ladhid and Ellyn Renou (Cherfils lab, CNRS and ENS Paris-Saclay) for their assistance with cloning.

## Author contributions

C.S., F.P., M.L., W.Z., and Y.F. performed the experimental studies. D.C., R.F., and Y.A. provided chemicals and plasmids. M.Z., S.C., Y.F., and J.C. carried out the analysis. M.-H.K. carried out the statistical analysis. Y.F. and J.C. supervised the work. J.C. wrote the manuscript.

## Competing interests

The authors declare no competing interests.
