## [Peer Review File · Nature Communications]

Membranes prime the RapGEF EPAC1 to transduce cAMP signalingREVIEWER COMMENTS

Reviewer #1 (Remarks to the Author):

The manuscript of Sartre et al. characterizes the critical and unexpected roles that membranes have in regulating the exchange activity of the EPAC1 Rap1 GEF. EPAC1 is an important component for cardiomyocyte function, and it is a potential therapeutic target. The manuscript is ground-breaking in several aspects:

1. It shows that anionic membranes, even in the absence of cAMP have an activating influence on the EPAC1 exchange activity, which the authors attribute to disrupting some autoinhibitory interactions upon membrane binding.
2. It presents X-ray crystallographic results that enable the authors to construct a plausible model for full-length EPAC1.
3. It uses HDX-MS to identify regions in the EPAC1 model that are likely to interact with lipid membranes, and the authors show that these interactions are dominated by basic residues on one face of EPAC1 interacting with membranes.
4. The authors propose four distinct structural states of EPAC1, dependent of cAMP binding and membrane binding. The authors make a convincing case that the extraordinary increase in cAMP affinity engendered by the membrane bound state of EPAC1 make this state suited as a target for inhibitor. This high affinity of the membrane-bound state of EPAC1 for cAMP dovetails nicely with the newly characterized cAMP nanodomains that have been described and suggest that bulk, soluble cAMP has a limited role in regulating EPAC1, and that EPAC1 in the cytosol would have negligible activity.
5. They show that an allosteric inhibitor of EPAC1 inhibits only the cAMP-bound state of the enzyme.

The manuscript is compact and clear. It is generally well written and easy to follow. Where there is speculation, it is clearly identified and appropriate.

There are only a few minor points that should be addressed:

1. Figure 1. The figure shows K_{cat} and K_{cat}/K_M . For panel C, the label should be PM liposomes. Not just liposomes. This is said in the legend, but once the authors make readers need to read the legend, they are more likely to lose them.
2. Fig 3D. The legend for the HDX figure is ambiguous. It states that it is "HDX-MS analysis of the activation of EPAC1 by cAMP and PM liposomes". However, it is not clear what is displayed. Is it 1. HDX on membrane - HDX in solution or 2. HDX in solution - HDX on membrane. This is important because it gives the sign of the difference to expect. I am guessing that since they say red means decreased exchange and the areas with decreased exchange map to the putative membrane-interacting surface, it should be HDX on the membrane - HDX in solution. The authors should say this explicitly.
3. The first place in the manuscript to use the name "switchboard" for a part of the structure is in the text on line 250, page 9. However, the "switchboard" is only illustrated in figure 5, which is first referred to on p. 12. The authors should explain what the switchboard is at the first use.
4. All HDX-MS results should be shown for each peptide in a supplementary file, i.e., for each peptide, a table should show HDX for all replicates, with the mean and SD for each peptide. In addition, the results should be deposited with the ProteomeXchange Consortium via the PRIDE partner repository.
5. Lines 324-325 say that anionic membranes "lower" the affinity for cAMP by two orders of magnitudes. I think this is a typo. It should be anionic membranes increase the affinity (lower the K_d).
6. The lipid used to form the membranes are not adequately described in the methods. They are just called PS, PE and PC. What are the fatty acid moieties? They came from Avanti, but what are the catalog numbers? These are important details, since the degree of saturation can be incredibly important for reproducing the results.

Reviewer #2 (Remarks to the Author):

Epac is a cAMP dependent Guanine Nucleotide Exchange Factor for the small G-protein Rap. The biophysical characterisation of the interaction between cAMP and Epac has shown that the affinity of Epac for cAMP is in the micromolar range. This finding is to some extent surprising as the affinity of Protein Kinase A (PKA) another well established receptor for cAMP is in the nanomolar range. This raised the question how sufficiently high cellular concentration of cAMP can be reached to activate Epac.

Here Sartre et al. demonstrate that the interaction of Epac with liposomes that functions as a model for membranes drastically increase the apparent affinity of Epac for cAMP. Furthermore, the mode of action of the Epac antagonist CE3F4 is analysed. To get a better understanding on the interaction between Epac and membranes, the crystal structure of the N-terminal half of Epac in complex with cAMP was solved. The presented data are timely and the experiments are well performed.

Major comments:

The authors claim two effects of membranes on the GEF activity of Epac. (i) an activating effect and (ii) an increased affinity for cyclic nucleotides. While the second effect is clear, more caution is required regarding the interpretation of the first effect. The authors compare the action of Epac on Rap in solution with the action of Epac on Rap in the presence of liposomes. In the latter case both Epac and Rap become attached to the surface of the liposome. Thus a situation in which the concentrations of Epac and Rap needs to be described in mol per volume is compared with a situation in which concentrations needs to be described in mol per surface area. It is therefore difficult to distinguish whether association of Epac to the liposomes is indeed increasing the activity (catalytic competence) of the Epac molecule or whether an observed increased in exchange activity is due to the fact that simply the probability of Epac to "meet" a Rap molecule is increases in the 2D situation, but without a change in the activity of the Epac molecule. This aspect should be discussed. It will not change the overall finding of the manuscript.

Minor comments:

The bargraphs in Fig. 2 showing apparent K_d values should be replace by graphs showing the original data (the dependency of the exchange activity on the cyclic nucleotide concentration). The data a currently partially presented as supplementary figures.

Reviewer #3 (Remarks to the Author):

Epac proteins are activated by binding of the second messenger cAMP and then act as guanine nucleotide exchange factors (GEFs) for Rap protein. The authors reported the GEF activation mechanism of EPAC1 induced by anionic membranes and cAMP and proposed an inhibition mechanism of CE3F4 by using structural and biochemical analyses. This work showed the influence and the importance of the membrane surface environment on the behavior of proteins. The result provides new insight into the regulation mechanism of EPAC1 and a way to develop a new drug for targeting EPAC1.

I described major/minor comments about this paper below.

Major points

(1) Overall, statistical analyses are necessary or evaluation criteria should be clearly described. For example, authors addressed that "addiction of cAMP did not increase binding of EPAC1(FL) to PA or CL-liposomes compared to neutral liposomes" (Line 144-145), but binding of EPAC1 seems to be increased in CL-liposome (+cAMP) (Fig. 1B).

(2) The authors determined the crystal structure of EPAC1(DEP)-EPAC1(CNB) in complex with cAMP and proposed that these two domains are rigidly associated with each other and behave as a single unit (Line211-). The mapping of HDX-MS analysis showed that the kinked helix region moves dynamically with membrane and cAMP. These two descriptions seem to be inconsistent.

Please discuss the rigidity of this helix.

(3) Several experiments were conducted using the different fragments, possibly making interpretation difficult.

Figure 4D showed CE3F4 inhibited the GEF activity for CNB-GEF fragment in the presence of cAMP, however SEC-SAXS and gel-filtration analysis were conducted using "kinked α -helix-CNB-GEF" fragment. Does the N-terminal kinked α -helix have any special roles in this analysis? In the fragment, kinked α -helix would be exposed, affecting the results.

The same fragments should be used for these experiments (Fig. 4B, 4C, 4D).

(4) SEC-SAXS analysis

(4-1) Authors should show all scattering curves, and add the error bar of the $P(r)$ function in Figure 4B.

(4-2) SEC-SAXS analysis revealed that the region of kinked α -helix-CNB-GEF extended its D_{max} by binding to the cAMP or CE3F4. However, the AMP alone stretched the molecule about 12~16Å longer than CE3F4 bound forms. The authors did not discuss this point well. If the authors revealed or proposed the molecular mechanism of this point, the paper would be better.

(4-3) What is a cause of a D_{max} extension? If cAMP or CE3F4 induces a significant domain re-arrangement, re-analysis of SAXS data may be effective/recommended.

It may be a way to calculate the relative position of domains using the program SASREF

(<https://www.emblhamburg.de/biosaxs/manuals/sasref.html>) and discuss the point.

The program Crysol (<https://www.emblhamburg.de/biosaxs/crysol.html>) may also be another way if the authors could make a good model.

Please try those.

(4-4) The dummy atom model calculation and presentation may be informative.

(4-5) The authors showed only R_g and I_0 plots of SEC-SAXS analysis in Fig. S6. The authors should add the A280 absorbance spectra of SEC-SAXS and compare them with the I_0 plot to evaluate whether both data are consistent. The plots around the peak would be sufficient, and other region could be removed. The labels of Y-axis should be shown. And, decimal points would be period.

Minor Points

(1) Line 325: "lowering the affinity" is right? "lowering the K_d -value" or "increasing the affinity" would be right.

(2) Electron density of cAMP should be shown.

(3) Figure 3C: Electron potential map would be better.

(3) In the method section of crystallographic analysis

Please describe the concentration of cAMP used in the crystallization experiment.

(4) In a method section of SAXS analysis

Did you use only one frame or a few frames?

If the authors prepared the scattering curve combined with a few frames, please write it.

(5) Period or comma are mixed as decimal points in the text and Table.

Line 509: 0,2

Table S1: Resolution 88.24-2,30

Rwork/Rfree 0.195/0,237

We are grateful to our reviewers for their positive comments and insightful suggestions, which have helped us greatly improve our manuscript. We have revised the manuscript accordingly and hope that we have answered all queries.

Our detailed responses are listed below and changes are highlighted in red in the manuscript.

Reviewer #1 (Remarks to the Author):

The manuscript of Sartre et al. characterizes the critical and unexpected roles that membranes have in regulating the exchange activity of the EPAC1 Rap1 GEF. EPAC1 is an important component for cardiomyocyte function, and it is a potential therapeutic target. The manuscript is ground-breaking in several aspects:

- 1. It shows that anionic membranes, even in the absence of cAMP have an activating influence on the EPAC1 exchange activity, which the authors attribute to disrupting some autoinhibitory interactions upon membrane binding.*
- 2. It presents X-ray crystallographic results that enable the authors to construct a plausible model for full-length EPAC1.*
- 3. It uses HDX-MS to identify regions in the EPAC1 model that are likely to interact with lipid membranes, and the authors show that these interactions are dominated by basic residues on one face of EPAC1 interacting with membranes.*
- 4. The authors propose four distinct structural states of EPAC1, dependent of cAMP binding and membrane binding. The authors make a convincing case that the extraordinary increase in cAMP affinity engendered by the membrane bound state of EPAC1 make this state suited as a target for inhibitor. This high affinity of the membrane-bound state of EPAC1 for cAMP dovetails nicely with the newly characterized cAMP nanodomains that have been described and suggest that bulk, soluble cAMP has a limited role in regulating EPAC1, and that EPAC1 in the cytosol would have negligible activity.*
- 5. They show that an allosteric inhibitor of EPAC1 inhibits only the cAMP-bound state of the enzyme.*

The manuscript is compact and clear. It is generally well written and easy to follow. Where there is speculation, it is clearly identified and appropriate.

There are only a few minor points that should be addressed:

- 1. Figure 1. The figure shows K_{cat} and K_{cat}/K_M . For panel C, the label should be PM liposomes. Not just liposomes. This is said in the legend, but once the authors make readers need to read the legend, they are more likely to lose them.*

• This has been done

- 2. Fig 3D. The legend for the HDX figure is ambiguous. It states that it is “HDX-MS analysis of the activation of EPAC1 by cAMP and PM liposomes”. However, it is not clear what is displayed. Is it 1. HDX on membrane – HDX in solution or 2. HDX in solution - HDX on membrane. This is important because it gives the sign of the difference to expect. I am guessing that since they say red means decreased exchange and the areas with decreased exchange map to the putative membrane-interacting surface, it should be HDX on the membrane – HDX in solution. The authors should say this explicitly.*

- We have clarified the legend as follows:
(HDX of EPAC1/cAMP/liposomes - HDX of EPAC1 in solution).

3. The first place in the manuscript to use the name “switchboard” for a part of the structure is in the text on line 250, page 9. However, the “switchboard” is only illustrated in figure 5, which is first referred to on p. 12. The authors should explain what the switchboard is at the first use.

- We have removed the name “switchboard” from the manuscript, and replaced it with GEF domain or hinge in the text and in the figures (Figures 5C and 5D). We also modified the description of the elements that respond to CE3F4 in the discussion as follows:
“We propose that this cAMP-responsive region is the hinge between the CNB and GEF domains, and that the inhibitory element is a flexible β -hairpin of the GEF domain located between the hinge and the Rap1-binding site”

4. All HDX-MS results should be shown for each peptide in a supplementary file, i.e., for each peptide, a table should show HDX for all replicates, with the mean and SD for each peptide. In addition, the results should be deposited with the ProteomeXchange Consortium via the PRIDE partner repository.

- We have added an extended description of the HDX-MS experiments and results in the Supplementary Information, including HDX for all replicates as requested. The results have been deposited via PRIDE database, but we apologize that they cannot be made public until a DOI is available.

5. Lines 324-325 say that anionic membranes “lower” the affinity for cAMP by two orders of magnitudes. I think this is a typo. It should be anionic membranes increase the affinity (lower the Kd).

- Thank you for pointing out this important typo to us. This has been corrected.

6. The lipid used to form the membranes are not adequately described in the methods. They are just called PS, PE and PC. What are the fatty acid moieties? They came from Avanti, but what are the catalog numbers? These are important details, since the degree of saturation can be incredibly important for reproducing the results.

- The catalog numbers of PE, PC, PS and PIP2 have been added to the methods. The fatty acid distribution for each lipid can be found at avantilipids.com

Reviewer #2 (Remarks to the Author):

Epac is a cAMP dependent Guanine Nucleotide Exchange Factor for the small G-protein Rap. The biophysical characterisation of the interaction between cAMP and Epac has shown that the affinity of Epac for cAMP is in the micromolar range. This finding is to some extent surprising as the affinity of Protein Kinase A (PKA) an other well established receptor for cAMP is in the nanomolar range. This raised the question how sufficiently high cellular concentration of cAMP can be reached to activate Epac.

Here Sartre et al. demonstrate that the interaction of Epac with liposomes that functions as a model for membranes drastically increase the apparent affinity of Epac for cAMP.

Furthermore, the mode of action of the Epac antagonist CE3F4 is analysed. To get a better understanding on the interaction between Epac and membranes, the crystal structure of the N-terminal half of Epac in complex with cAMP was solved. The presented data are timely and the experiments are well performed.

Major comments:

The authors claim two effects of membranes on the GEF activity of Epac. (i) an activating effect and (ii) an increased affinity for cyclic nucleotides. While the second effect is clear, more caution is required regarding the interpretation of the first effect. The authors compare the action of Epac on Rap in solution with the action of Epac on Rap in the presence of liposomes. In the latter case both Epac and Rap become attached to the surface of the liposome. Thus a situation in which the concentrations of Epac and Rap needs to be described in mol per volume is compared with a situation in which concentrations needs to be described in mol per surface area. It is therefore difficult to distinguish whether association of Epac to the liposomes is indeed increasing the activity (catalytic competence) of the Epac molecule or whether an observed increased in exchange activity is due to the fact that simply the probability of Epac to “meet” a Rap molecule is increases in the 2D situation, but without a change in the activity of the Epac molecule. This aspect should be discussed. It will not change the overall finding of the manuscript.

- We agree with this reviewer that the probability for EPAC1 to “meet” a Rap GTPase may be increased on the membrane and that this could contribute to its GEF efficiency on the membrane.

However, we and others observed that apo-EPAC1 has no measurable GEF activity towards Rap1 in solution, where it is fully autoinhibited. Autoinhibition involves the kinked helix that connects the DEP and CNB domain in the regulatory N-terminus, as shown by the fact that a CNB-GEF construct that includes this helix is autoinhibited in solution in the absence of cAMP, while a CNB-GEF construct that lacks the helix is partially active under these conditions (this is shown in Figure S5D). Thus, to stimulate nucleotide exchange on Rap GTPases, EPAC1 must be activated by displacement of its regulatory N-terminus to relieve autoinhibition. Since EPAC1 displays a GEF activity on membranes in the absence of cAMP, it can be deduced that membranes promote such an activating conformational change, hence that membranes change the activity of EPAC1.

To highlight that apo-EPAC1 is inactive in solution, we have added the corresponding curve to panel S1D.

To discuss the possible contribution of membranes to EPAC1 efficiency through reduction of dimensionality from 3D to 2D, we have added the following sentence in the discussion: “In addition to this direct activating effect, localization of EPAC1 and Rap GTPases at the membrane surface may increase their encounter probability through dimensional reduction, which may add to the GEF efficiency of EPAC1 on membranes.”

Minor comments:

The bargraphs in Fig. 2 showing apparent K_d values should be replace by graphs showing the original data (the dependency of the exchange activity on the cyclic nucleotide concentration). The data a currently partially presented as supplementary figures.

- We have merged Figure 2 and S2, which now includes both the original data (previously in Figure S2) and the bar graphs. The supplementary Figures S3-S7 have been renumbered S2-S6 accordingly.

Reviewer #3 (Remarks to the Author):

Epac proteins are activated by binding of the second messenger cAMP and then act as guanine nucleotide exchange factors (GEFs) for Rap protein. The authors reported the GEF activation mechanism of EPAC1 induced by anionic membranes and cAMP and proposed an inhibition mechanism of CE3F4 by using structural and biochemical analyses. This work showed the influence and the importance of the membrane surface environment on the behavior of proteins. The result provides new insight into the regulation mechanism of EPAC1 and a way to develop a new drug for targeting EPAC1. I described major/minor comments about this paper below.

Major points

(1) Overall, statistical analyses are necessary or evaluation criteria should be clearly described. For example, authors addressed that “addition of cAMP did not increase binding of EPAC1(FL) to PA or CL-liposomes compared to neutral liposomes” (Line 144-145), but binding of EPAC1 seems to be increased in CL-liposome (+cAMP) (Fig. 1B).

Statistical analysis have been added to all figures (1A, 1B, 1C, 1D, 1E, 2B, 2C, 2E, 2F, 3A, 3D, 3E, 3F). The statistical analyses are described in a new section of the methods and are summarized in a new supplementary Table S3. The statistical analyses do not modify the previous conclusions of our manuscript, including that of Figure 1B. .

(2) The authors determined the crystal structure of EPAC1(DEP)-EPAC1(CNB) in complex with cAMP and proposed that these two domains are rigidly associated with each other and behave as a single unit (Line211-). The mapping of HDX-MS analysis showed that the kinked helix region moves dynamically with membrane and cAMP. These two descriptions seem to be inconsistent. Please discuss the rigidity of this helix.

- Our crystallographic analysis suggests that the kinked helix is displaced upon activation as part of the DEP-helix-CNB rigid block. We clarified this in the crystallography section as follows :

“Accordingly, the kinked helix, which forms autoinhibitory interactions with the GEF domain (see below, **Figure S4D**) is predicted to be displaced upon activation as part of the rigid DEP-CNB unit.”

The HDX-MS data showing changes at this helix upon activation are consistent with 1) the structure of autoinhibited EPAC2, which shows that the helix blocks the Rap GTPase-binding site (Rehmann 2006, Ref 19), hence must be displaced in active EPAC1 2) our biochemical data showing that the helix-CNB-GEF construct is autoinhibited in solution, while the CNB-GEF construct has partial activity (Figure S4D, as also reported in Kraemer 2001, ref 22), confirming that the helix is autoinhibitory in EPAC1. Thus, the HDX-MS data do not contradict our crystallographic analysis. To make that point clearer, we expanded the description of the biochemical experiment showing that the kinked helix is autoinhibitory as follows:

“In solution, we used EPAC1^{CNB-GEF}, a construct that lacks the helix located between the DEP and CNB domain which is predicted to form autoinhibitory interactions with the GEF domain in the absence of cAMP. EPAC1^{CNB-GEF} displays intrinsic GEF activity in the absence of cAMP, confirming that removal of the helix results in partial release of autoinhibition, and this intrinsic activity is increased by cAMP (Figure S4D, ²²).”

(3) Several experiments were conducted using the different fragments, possibly making interpretation difficult.

Figure 4D showed CE3F4 inhibited the GEF activity for CNB-GEF fragment in the presence of cAMP, however SEC-SAXS and gel-filtration analysis were conducted using “kinked α -helix-CNB-GEF” fragment. Does the N-terminal kinked α -helix have any special roles in this analysis? In the fragment, kinked α -helix would be exposed, affecting the results.

The same fragments should be used for these experiments (Fig. 4B, 4C, 4D).

- We used the CNB-GEF construct that lacks the kinked helix in the biochemical experiment reported in Figure 4D because this construct displays intrinsic GEF activity in solution while remaining sensitive to activation by cAMP. This property allowed us to analyze whether CE3F4 requires cAMP for inhibition in solution, which would not be possible with the kinked helix-CNB-GEF construct which is entirely inactive in solution.

(4) SEC-SAXS analysis

(4-1) Authors should show all scattering curves, and add the error bar of the $P(r)$ function in Figure 4B.

- The error bars have been added to the $P(r)$ function in Figure 4B.
- The scattering curves are now shown in Figure S5F.

(4-2) SEC-SAXS analysis revealed that the region of kinked α -helix-CNB-GEF extended its D_{max} by binding to the cAMP or CE3F4. However, the AMP alone stretched the molecule about 12~16Å longer than CE3F4 bound forms. The authors did not discuss this point well. If the authors revealed or proposed the molecular mechanism of this point, the paper would be better.

(4-3) What is a cause of a D_{max} extension? If cAMP or CE3F4 induces a significant domain re-arrangement, re-analysis of SAXS data may be effective/recommended.

It may be a way to calculate the relative position of domains using the program SASREF (<https://www.emblhamburg.de/biosaxs/manuals/sasref.html>) and discuss the point.

The program Crysol (<https://www.emblhamburg.de/biosaxs/crysol.html>) may also be another way if the authors could make a good model. Please try those.

(4-4) The dummy atom model calculation and presentation may be informative.

- As summarized in Figure 5A, our data suggest that EPAC1 visits different conformational states, defining conformational ensembles, and that the equilibrium between these states is shifted by membranes and cAMP. Accordingly, as discussed in the text, each SEC-SAXS dataset is representative of a distinct conformational ensemble. The differences observed upon addition of CE3F4 thus reveal that binding of CE3F4 alters the conformational ensemble. However, we prefer to refrain from modeling individual conformations which together would account for the observed SAXS data, as to our opinion the conformational ensembles would be underdetermined (fewer observations than unknowns).

To make these points clearer, we have added “The conformational ensemble of EPAC1” to the subtitle of Figure 5A and we have added double arrows to the corresponding Figure 5A to indicate that the different states are in equilibrium.

(4-5) The authors showed only R_g and I_0 plots of SEC-SAXS analysis in Fig. S6. The authors should add the A280 absorbance spectra of SEC-SAXS and compare them with the I_0 plot to evaluate whether both data are consistent. The plots around the peak would be sufficient, and other region could be removed. The labels of Y-axis should be shown. And, decimal points would be period.

- We have added a new panel S5E showing the A280 absorbance spectra for the four different SEC-SAXS experiments.
- Panels S5A-D have been improved as suggested.

Minor Points

(1) Line 325: “lowering the affinity” is right? “lowering the K_d -value” or “increasing the affinity” would be right.

- Thank you for alerting us on this important typo. “lowering the affinity” has been replaced by “increasing the affinity”.

(2) Electron density of cAMP should be shown.

- We have added a new panel in Figure S2B showing the electron density (omit map) of cAMP.

(3) Figure 3C: Electron potential map would be better.

We agree that the electron potential map could be an alternative representation. However, we have preferred to keep the current representation in which the positively charged residues can be more clearly located on individual domains.

(3) In the method section of crystallographic analysis

Please describe the concentration of cAMP used in the crystallization experiment.

- The concentration is 10 mM. It has been added in the method section.

(4) In a method section of SAXS analysis

Did you use only one frame or a few frames?

If the authors prepared the scattering curve combined with a few frames, please write it.

- We already indicated in the submitted manuscript that “Frames corresponding to the high-intensity fractions of the peak and having constant radius of gyration (R_g) within error were averaged”. We have now added the frame numbers in the legend of Table S2.

(5) Period or comma are mixed as decimal points in the text and Table.

Line 509: 0,2

Table S1: Resolution 88.24-2,30
Rwork/Rfree 0.195/0,237

- Thank you for alerting us on these typos. Decimal points have been changed to periods throughout the manuscript.
- Other Changes:
 - To acknowledge the contribution of Marie-Hélène Kryszke in the statistical analysis added to the revised version, this author was moved from 6th to 4th author. Gérald Peyroche, previously second to last author, has been removed from the co-authors on his request. To acknowledge the contribution of Yann Ferrandez as a senior co-author, this author was moved from second co-author to second to last co-author.
The final ordered list of authors is now as follows:
Candice Sartre, François Peurois, Marie Ley, Marie-Hélène Kryszke, Wenhua Zhang, Delphine Courilleau, Rodolphe Fischmeister, Yves Ambroise, Mahel Zeghouf, Sarah Cianferani, Yann Ferrandez, Jacqueline Cherfils
- The abstract has been reduced to less than 150 words
- All figures have been redrawn to include all experimental points.
- Author contributions have been added
- Competing interests have been added

REVIEWERS' COMMENTS

Reviewer #3 (Remarks to the Author):

All of my concerns are addressed. I have no further questions and I accept this manuscript for publication.